# White Box Network: Obtaining a right composition ordering of functions

## Abstract

Neural networks have significantly benefitted real-world tasks. The universality of a neural network enables the approximation of any type of continuous functions. However, a neural network is regarded as a non-interpretable black box model, and this is fatal to reverse engineering as the main goal of reverse engineering is to reveal the structure or design of a target function instead of approximating it. Therefore, we propose a new type of a function constructing network, called the white box network. This network arranges function blocks to construct a target function to reveal its design. The network uses discretized layers, thus rendering the model interpretable without disordering the function blocks. Additionally, we introduce an end-to-end PathNet structure through this discretization by considering the function blocks as neural networks

## 1 Introduction

*Reverse engineering* can be defined as the deconstruction of an object to reveal its design or architecture, or to extract knowledge from the object [Eldad, 2005]. Typically, in machine learning, obtaining the internal functions of a device only with its input and output values is a regression problem. Before machine learning was developed, devices were disassembled to analyze them. However, advances in deep learning have allowed us to approximate various functions in regression tasks; therefore, we can now solve this regression problem without actually looking inside the object.

While this development in deep learning (Bengio, 2009; Bengio et al., 2013) has benefitted reverse engineering significantly, some limitations still exist. First, deep learning is a black box model; therefore, the results may not be interpreted. In this case, information regarding the object such as what it does and how it works may not be extracted. This causes field engineers to not entirely trust the results of deep learning even if the correct values were reflected. Next, deep learning approximates the target function and does not reveal the actual design and architecture of the device. An artificial neural network is a mathematical simulation of a neural network composed of human neurons and synapses that simulate the manner in which the human brain recognizes patterns. However, a device typically comprises an electronic circuit and block functions; therefore, it may not fit perfectly with the neural network structure, which can cause large errors when certain data are used. We herein propose a white box network (WBN) with a selection layer, specially designed for reverse engineering. This network continuously composites function blocks to create a target function. We assume that all types of function blocks that can constitute the target function are provided in advance. This differs from the normal neural network because the WBN reveals the exact functions with the correct inputs and their ordering to construct the target function, instead of merely approximating them. Our WBN is different from normal neural networks as the model architecture is discretized using a discretized matrix called the selection layer. Because discretization is interpretable (Chen et al., 2016), our WBN can reconstruct the target function.

The WBN is motivated by the questions, Can we benefit from knowing the functions that are used in certain fields?. For example, in a programmable logic controller (PLC), a target function can be obtained through a ladder logic diagram comprising well-known function blocks. Therefore, we can obtain a target function by ordering those function blocks with the correct inputs. As such, in this study, we create a WBN that can automate reverse engineering. For the experiment, we imitated the PLC data and verified whether our model could obtain their ladder logic diagram.

Additionally, we conducted other experiments to validate the selection layer. We assumed that we

may not know the functions used in a certain field. Therefore, we regarded the given function blocks as neural networks. Subsequently, the network was ordered and connected to the appropriate neural networks to obtain the target function that resembles PathNet (Fernando et al., 2017). Using this concept, we create an end-to-end PathNet structure and perform experiments to verify whether it functions like PathNet.

## 2 RELATED WORKS

In machine learning, regression is a task of obtaining a real-valued function $y = f(x)$ from the hypothesis set $(X, Y)$. However, typically, actual data may contain noise or exhibit data loss owing to measurement limitations or other causes. Therefore, regression is often considered as a black box process and an exact function may not be obtained. For example, in Gaussian process regression [Williams and Ras-mussen, 2006] and support vector regression (SVR) (Smola & Schölkopf, 2004), we reproduced a kernel and obtained an approximated value instead of obtaining a function. Another typical regression method is using a multilayered neural network (Bengio, 2009; Bengio et al., 2013), which approximates a function by using the human neuron model.

The aim of reverse engineering is to obtain the exact structure of a function rather than merely approximating it. In some fields, we may know the type of function forms that typically appear; hence, we can use that information to construct a function. An equation learner (EQL) (Martius & Lampert, 2017; Sahoo et al., 2018) is designed for regression task that can be described by analytical expressions of mechanical systems such as a pendulum or a robotic arm. An EQL has been shown more effective at solving dynamics than conventional neural nets; additionally, it reduces extrapolation errors. Furthermore, PDE-net (Long et al., 2018b;a) accurately predicts the dynamics of complex systems and uncovers the underlying hidden partial differential equation models using a convolution kernel filter. This type of model renders regression more effective if certain types of function blocks or processes that appear in a specific field are known.

A discrete variable can be used variously, e.g., learning probabilistic latent representations (Kingma et al., 2014), image region (Xu et al., 2015), and memory access (Graves et al., 2014; 2016). Especially for the memory access, it resembles an attention mechanism that can help interpret the focus of the model. The end-to-end memory network (Kingma et al., 2014) uses softmax to access the memory in questions and answering task. This allows us to determine the sentence that the network refers to when it answers a specific question. The neural Turing machine (NTM) (Graves et al., 2014) and differentiable neural computer (Graves et al., 2016) use a sharpening method instead of merely using softmax. As such, those models select information from the memory without disordering information, thus rendering them more interpretable. Therefore, we can determine how the model functions. For example, we can reconstruct the Copy processes of the NTM and create a pseudocode by referring to the headers in the model. Therefore, discretization can be used such that the model is more interpretable (Chen et al., 2016), and it is sometimes more efficient for computing (Rae et al., 2016).

PathNet (Fernando et al., 2017) is a modular deep neural network with pathways. It uses a genetic algorithm to obtain the best pathway for learning. The pathways are represented as integer-valued matrices; therefore, the modules can be selected by accessing those matrices. PathNet reuses the pathways for a different task as a transfer learning paradigm and shows the potential in selecting the appropriate modules to obtain the target results.

## 3 METHODS

### 3.1 WHITE BOX NETWORK ARCHITECTURE

The WBN is a multilayered feed forward network. For each hidden layer, it consists of a selection layer followed by function block mappings. In this study, we assumed that all function blocks constituting the target function were known before we built the model. However, the exact order of the function blocks with the correct inputs must be obtained. Each function block can be multivalued with multiple inputs; however, for simplicity, we assume that the function blocks are unitary and binary operations.

Suppose that the model is $L$-layered, and we have $m(l)$ unitary functions and $n(l)$ binary functions in the $l$th layer. In the selection layer, we select inputs to enter into each function block. The

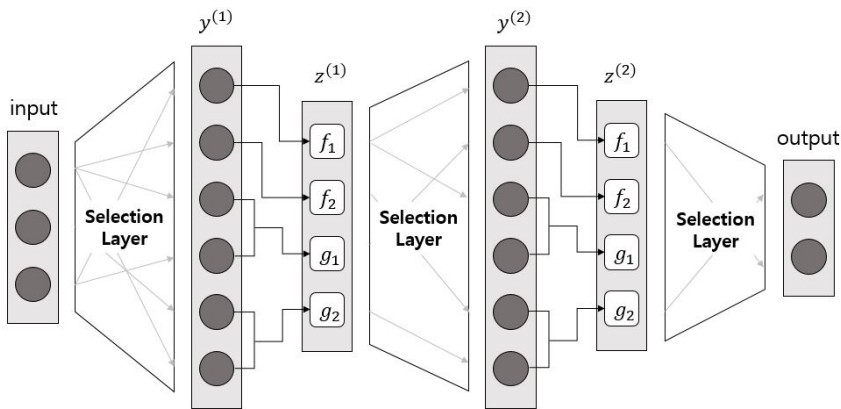

Figure 1: Architecture of white box network for two layers ($L = 2$) with each layer containing two unitary operations $\{f_1, f_2\}$ and two binary operations $\{g_1, g_2\}$ as its function blocks.

selection layer can be written as

$$z^{(l)} = \hat{W}^{(l)} y^{(l-1)}$$

Here, $y^{(l-1)}$ is the output of the previous layer and $\hat{W}^{(l)}$ is the selection matrix. If each row of $\hat{W}^{(l)}$ has only one 1 and the remaining 0s, then, it selects one element of the $y^{(l-1)}$ as an element of $z^{(l)}$. To make $\hat{W}^{(l)}$ differentiable, we approximate $\hat{W}^{(l)}$ by

$$\hat{W}^{(l)} = \text{softmax}(W^{(l)}/\tau, \text{ axis=-1})$$

Here, $\tau > 0$ is a temperature parameter that renders $\hat{W}$ sharper as $\tau \to 0$. $\tau$ can be chosen as both a constant and a learnable parameter.

After multiplying the selection matrix, $z^{(l)}$ goes through the function block mappings. Let $f_1^{(l)}, \ldots, f_{m(l)}^{(l)}$ and $g_1^{(l)}, \ldots, g_{n(l)}^{(l)}$ be the given unitary and binary function blocks on the $l$th layer. Then, our new output $y^{(l)}$ is

$$y^{(l)} = \left( f_1^{(l)}(z_1^{(l)}), \ldots, f_{m(l)}^{(l)}(z_{m(l)}^{(l)}), g_1^{(l)}(z_{m(l)+1}^{(l)}, z_{m(l)+2}^{(l)}), \ldots, g_{n(l)}^{(l)}(z_{m(l)+2n(l)-1}^{(l)}, z_{m(l)+2n(l)}^{(l)}) \right)$$

As such, when we select the function blocks of each layer, the dimensions of the variables are fixed. For example, $y^{(l)}$ represents $(m(l) + n(l))$-dimensional vectors and $z^{(l)}$ represents $(m(l) + 2n(l))$-dimensional vectors.

Finally, we convert the last layer's output to fit the target output dimension. Therefore, we use the selection layer again to match with the output dimension.

$$o = \hat{W}^{(L+1)} y^{(L)}$$

Here, the final output is a composition of function blocks from each layer with selective inputs.

## 3.2 END-TO-END PATHNET ARCHITECTURE

Suppose we do not have any information regarding the target function. In that case, we can select our function blocks as a single-layered neural network followed by nonlinear activation. Then, our model becomes modular neural network that selects the right modules to find the best pathway to obtain the target value. This concept is the same as that of a previous study, i.e., PathNet (Fernando et al., 2017).

To apply one key component of our WBN to existing PathNet architecture, we need to switch the order of the selection layer and the function block mapping for each hidden layer. Therefore, function block mapping is proceeded by the selection layer. In function block mapping, the function blocks are one-layered neural networks with learnable parameters. Each of them is assigned a single input and yields a single output, which is typically multidimensional.

Now, suppose that our model is a modular deep neural network with $L$ layers with each layer containing $M$ neural network modules. Let $N$ be the number of maximum distinct modules per layer

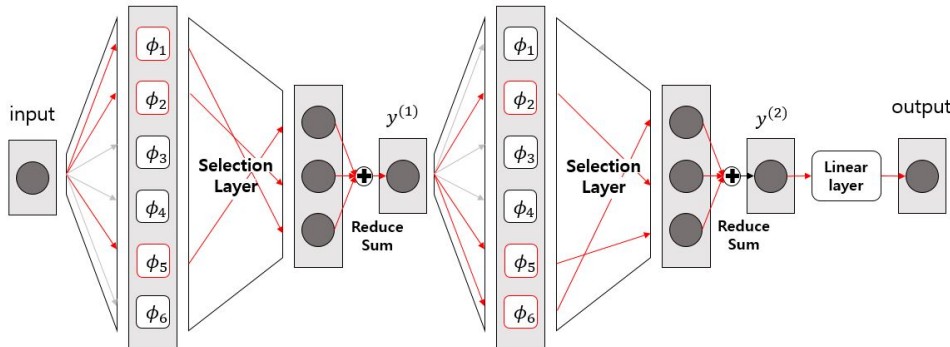

Figure 2: Architecture of end-to-end PathNet comprising a two-layer ($L = 2$) network with six modules ($M = 6$) in each layer. The number of maximum distinct modules per layer that are permitted in the pathway is three ($N = 3$), and the final layer comprises a fully connected linear layer. The activated path and modules are colored in red.

that can be selected in a pathway.

Let $\phi_1^{(l)}, , \phi_M^{(l)}$ be neural network modules in the $l$th layer and let $k^{(l)}$ be the dimension(number of channels) of hidden output variables. Then, the function mapping layer is written as

$$\boldsymbol{Z}^{(l)} = (\phi_1^{(l)}(\boldsymbol{y}^{(l-1)}), \phi_2^{(l)}(\boldsymbol{y}^{(l-1)}), \dots, \phi_M^{(l)}(\boldsymbol{y}^{(l-1)}))$$

Here, $\boldsymbol{Z}^{(l)}$ is $M$ by $k^{(l)}$ matrix. Then, selection layer selects the neural network outputs

$$\hat{\boldsymbol{Y}}^{(l)} = \hat{\boldsymbol{W}}^{(l)} \boldsymbol{Z}^{(l)}$$

The selection matrix $\hat{\boldsymbol{W}}^{(l)}$ is $N \times M$ with each row of $\hat{\boldsymbol{W}}^{(l)}$ ) containing only one almost 1 and the remaining almost 0s; $\hat{\boldsymbol{W}}^{(l)}$ can be approximated in the manner described previously. Now, we add all rows of $\hat{\boldsymbol{Y}}^{(l)}$ such that the hidden output $\boldsymbol{y}^{(l)}$ follows the existing PathNet.

$$\boldsymbol{y}^{(l)} = \text{reduce\_sum}(\hat{\boldsymbol{Y}}^{(l)}, \text{ axis=0})$$

Finally,there is a linear layer that can be used to convert the last hidden layer to the output value.

$$\boldsymbol{o} = \boldsymbol{W}\boldsymbol{y}^{(L)} + \boldsymbol{b}$$

This architecture is inspired by the PathNet in which a selection matrix instead of a genetic algorithm is used to select a pathway. It is differentiable, and hence we can easily train the network by gradient descent.

## 3.3 TRAINING DETAILS

We used the stochastic gradient descent algorithm with mini-batches and Adam (Kingma & Ba, 2015) to update the parameters. The free parameter of the WBN is $\theta = \{\boldsymbol{W}^{(1)}, \dots, \boldsymbol{W}^{(L)}\}$ or possibly $\theta = \{\boldsymbol{W}^{(1)}, \dots, \boldsymbol{W}^{(L)}, \tau^{(1)}, \dots, \tau^{(L)}\}$. Furthermore, we used the mean square error as a loss function for the regression task and an $L_1$ regularization term for $\boldsymbol{W}^{(l)}$'s

$$\text{Loss}(D) = \frac{1}{|D|} \sum_{i=1}^{|D|} ||\phi(\boldsymbol{x}_i) - \boldsymbol{y}_i||^2 + \alpha \sum_{i=1}^{L} |\boldsymbol{W}^{(l)}|_1$$

which is a Lasso-like (Tibshirani, 1996). For the WBN, $L_1$ regularization is highly important in optimization. Without the regularization term, the absolute value of elements of $\boldsymbol{W}^{(l)}$ can become extremely large. This causes $\hat{\boldsymbol{W}}^{(l)}$ to be extremely sparse and sharp, and it can be an obstacle for shifting the function blocks from one ordering to another. As expected, the model typically fails to obtain the exact ordering of function blocks when we set $\alpha = 0$. Hence, the regularization parameter $\alpha$ should be set to more than 0. Typically, we set $\alpha \in (0, 5]$ at the beginning of training to prevent the problem.

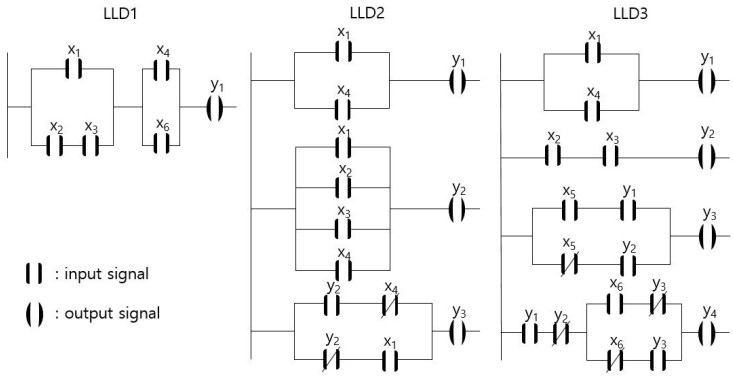

Figure 3: Three ladder logic diagrams (LLDs) for the experiment.

However, another problem occurs with the scale of $\alpha$. We typically set $\alpha$ to be more than 0 at the beginning; however, after the model has discovered the exact ordering, $\hat{W}^{(l)}$ cannot be sharpened enough because of the large regularization restriction. Therefore, we change the regularization parameter $\alpha$ during training time; we set $\alpha = 0$ after enough learning time. Two methods exist for selecting when to set $\alpha = 0$: one is by a threshold and the other is by an iteration number. For the former, if the MSE is smaller than a certain threshold (typically 0.001), we set $\alpha = 0$. For the latter, if the iteration reaches a certain number (typically $t = T/2$, where T is the total number of iterations), we set $\alpha = 0$.

Occasionally, we used the temperature learning strategy such that the temperature of each layer can be learned during the optimization. We used $\tau (> 0)$ because we wished to sharpen our selection matrix $\hat{W}^{(l)}$; however if $\tau$ is extremely small at the early stage of training, the model may be prevented from selecting the exact order of function blocks. Because we want our $\tau$ to approach zero as learning progresses, we use

$$\hat{\tau}^{(l)} = \frac{1}{\tau_0 + \text{softplus}(\tau^{(l)})} \quad \text{or} \quad \hat{\tau}^{(l)} = \frac{1}{\exp(\tau_0 + \text{softplus}(\tau^{(l)}))}$$

$$\hat{W}^{(l)} = \text{softmax}(W^{(l)}/\hat{\tau}^{(l)}, \text{ axis=-1})$$

where $\tau_0$ is a constant determining upper bound of $\hat{\tau}$.

## 4 EXPRIMENTS

### 4.1 LADDER LOGIC DIAGRAM

The ladder logic diagram (LLD) is a programming language that represents a program by a graphical diagram based on the circuit diagrams of a relay logic hardware. It is especially used in developing software for PLCs. Typically, a PLC device is black boxed before it is released as that is the core part of the technology. In fact, an attempt to model a PLC using recurrent neural networks has been reported (Abdelhameed & Darabi, 2005). However, the tracking and analysis of the learning process were inadequate; therefore, the internal structure of the controller could not be determined. Our white box model is designed to reveal this internal structure, and our goal in this experiment is to restore the logic diagram through reverse engineering using the input and output signals of a PLC device.

Three LLDs were used in our experiment, as shown in Figure 3. These LLDs range from simple to complex. The first one is a single output problem and the last two are multiple output problems. Furthermore, in the last two diagrams, the output signals become input signals that create other outputs; therefore, the outputs are highly related to each other. The maximum number of inputs required was six, but we set the input signals as eight dimensional to validate whether the network could find the LLDs even if nonmeaningful input values existed.

The inputs and outputs are Boolean numbers, which are either 0 or 1. Therefore, the function

Table 1: Mean squared errors for ladder logic diagrams

|  | LLD1 | LLD2 | LLD3 |
|---|---|---|---|
| WBN | $(7.404 \pm 0.166) \times 10^{-7}$ | $(2.425 \pm 0.160) \times 10^{-6}$ | $(3.312 \pm 0.108) \times 10^{-6}$ |
| ANN | $(2.729 \pm 0.287) \times 10^{-6}$ | $(7.422 \pm 0.280) \times 10^{-5}$ | $(9.619 \pm 0.216) \times 10^{-5}$ |
| SVR | $1.181 \times 10^{-2}$ | $1.861 \times 10^{-2}$ | $7.123 \times 10^{-1}$ |

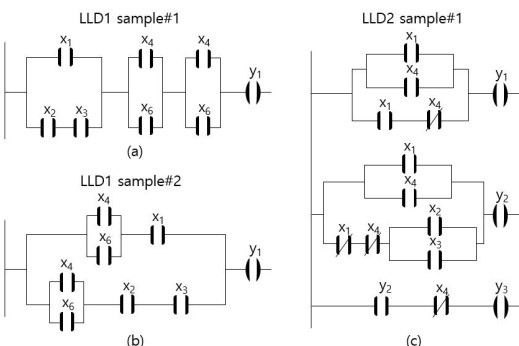

Figure 4: Sample structures for three LLDs suggested by WBN. (a) LLD1 sample #1 constructed by WBN (b) LLD1 sample #2 constructed by WBN (c) LLD2 sample #1 constructed by WBN

blocks used in LLDs are nondifferentiable Boolean discrete functions. To solve this problem, we continuously extended the function blocks such that gradient descent could be applied. The four function blocks used in this experiment are as follows.

$$\text{IDENTITY}(x) = x$$
$$\text{NOT}(x) = 1 - x$$
$$\text{AND}(x, y) = x \times y$$
$$\text{OR}(x, y) = x + y - x \times y$$

The first two are unitary operations and the last two are binary. Therefore, the number of unitary functions $m = 2$ and the number of binary functions $n = 2$. We used these functions multiple times by duplicating them. Hence, the number of functions was set to $m + n \in \{12, 16, 20\}$ and the number of layers $L \in \{3, 4, 5, 6\}$ was selected in our experiment. The initial $L_1$ regularization parameter $\alpha$ was set to $\alpha \in \{0.1, 0.5, 1, 5\}$.

We prepared a multilayered artificial neural network (ANN) and support vector regression (SVR) to compare with our model. For the ANN, the number of layers were set as that of our model and the number of hidden units $k \in \{5, 10, 20\}$ per layer. For SVR, we used a radial basis function kernel of width $\gamma \in \{0.05, 0.1, 0.5, 1.0\}$ and set two hyperparameters $C \in \{0.01, 0.1, 1, 10, 100\}$ and $\epsilon \in \{0.01, 0.1, 1\}$. Although comparing losses is not meaningful as our model not only reduces the error, but also obtains the actual structure of the target function, it clearly shows how our model effectively reduces the loss after obtaining the structure of the target function.

Three samples of LLDs suggested by our model are shown in Figure 4. (a) is a sample from LLD1, which is equivalent to LLD1. Two equivalent LLDs imply that, for every possible input signal, the corresponding output signals from two LLDs are the same. Therefore, we verified that the WBN may not suggest the exact diagram but an equivalent diagram (it sometimes produces the exact LLD). (b) is another sample of LLD1 that is also equivalent to LLD1. We discovered that if we set the number of layers larger than the required number, the WBN will likely generate an inefficient LLD. (b) is a sample of LLD2; it is equivalent to LLD2. Both $y_1$ and $y_2$ cannot be obtained efficiently using this diagram. However, it is clear that $y_3$ is of a simpler form than the original LLD2; additionally, it is equivalent to real $y_3$ in LLD2 because $\text{NOT}(y_2)$ and $x_1$ cannot occur simultaneously. We included more samples of LLDs including samples from LLD3 in the Appendix. See Figure 7

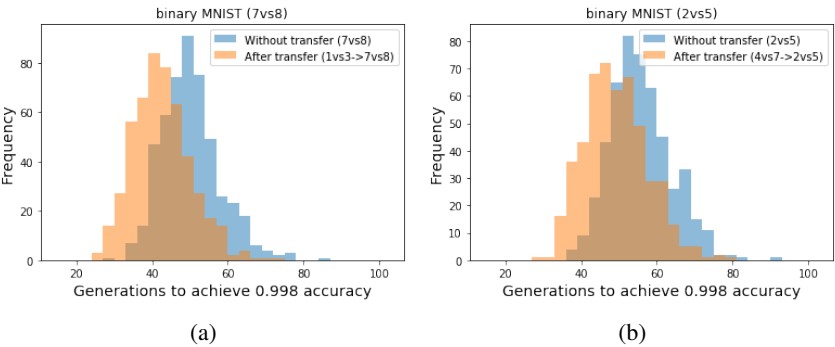

Figure 5: Binary MNIST classification results. (a) transfer result for 1vs3 → 7vs8. Total speed-up radio is 1.18. (b) transfer result for 4vs7. → 2vs5. Total speed-up radio is 1.13

## 4.2 BINARY MNIST CLASSIFICATION

We performed experiments to verify the end-to-end PathNet structure. We used the selection matrix instead of the genetic algorithm, which is different from the original PathNet. Experiments were conducted to determine whether the selection matrix could select the appropriate path. We performed the experiments using the transfer learning paradigm suggested in the original PathNet and observed a positive transfer by using our network.

A binary MNIST classification is a task distinguishing two classes of MNIST digits from each other. In our experiment, we conducted two tasks for the transfer learning paradigm. First, we trained binary classification task A until we obtained a 0.998 training accuracy. After we achieved the accuracy, we set the best-fit pathway, that is, the parameters within the pathway were no longer allowed to change. Subsequently, we initialized the remaining parameters including the last linear layer and trained another binary classification task B until we obtained a 0.998 training accuracy.

In this experiment, we adopted the original PathNet structure to construct our model. The end-to-end PathNet structure comprises $L = 3$ layers; each layer contains $M = 10$ linear units with 20 neurons each followed by rectified linear units. We set $N = 3$ as the maximum number of distinct modules permissible in a pathway for each layer.

We created two different datasets for transfer learning. One is for transferring from classifying 1 and 3 to classifying 7 and 8 (1vs3 to 7vs8). The other is for transferring from classifying 4 and 7 to classifying 2 and 5 (4vs7 to 2vs5). We created these datasets because, in our opinion, transferring 1vs3 to 7vs8 is more reasonable than transferring 4vs7 to 2vs5 because 1, 4, 7 are numbers with only straight lines, whereas 2, 5 , 8 are numbers with curved lines. We expect that the model that is learned to classify 1 and 3 can classify 7 and 8 easily after a transfer. The experiment was performed 500 times for each dataset to analyze the statistics.

Figure 5 shows the results of a positive transfer. For classifying 7 and 8, the number of generations to obtain a 0.998 training accuracy was 49.728 and 42.112 on average before and after transfer, respectively. The speed-up ratio was 1.18 for this dataset. For classifying 2 and 5, the number of generations to obtain a 0.998 training accuracy was 55.324 and 48.808 on average before and after transfer, respectively. The speed-up ratio was 1.13 for this dataset. We discovered that the learning speed of the task enhanced for both datasets. Furthermore, it is clear that the transfer from 1vs3 to 7vs8 is better than that of the other by comparing the speed-up ratios. This implies that the selection matrix successfully discovered the best way pathway to transfer. The loss decreases quickly at the early stage of the training for the post-transferred task compared with the nontransferred task. See Figure 9 in appendix.

## 4.3 CIFAR CLASSIFICATION

The CIFAR10 dataset contains images from 10 different classes. We performed an experiment using this dataset to determine whether our network could be used for classification problems. In the original PathNet study, a similar experiment was performed, in which information was transferred between CIFAR and SVHN datasets. One is for the classification of 10 different objects and the other that of 10 different digits. In our opinion, using two dataset are not suitable for transfer learning as

they need to focus on different items when classifying them. Therefore, we divided the CIFAR dataset into two classes: class 1, which comprises data from airplanes, automobiles, birds, cats, and deer; and class 2, which comprises data from dogs, frogs, horses, ships, and trucks.

In our experiment, we used a transfer paradigm similar to that of the previous experiment. However, we trained the task according to a fixed iteration number instead of obtaining a certain accuracy. We trained one class 5000 times and then fixed the best-fit pathway. Subsequently, we initialized the remaining parameters and trained the other class 5000 times to verify the accuracy with their test sets.

The end-to-end PathNet structure we used comprises $L = 3$ layers; each layer contains $M = 16$ linear units with 20 neurons each followed by rectified linear units. We set $N = 4$ as the maximum number of distinct modules permissible in a pathway for each layer.

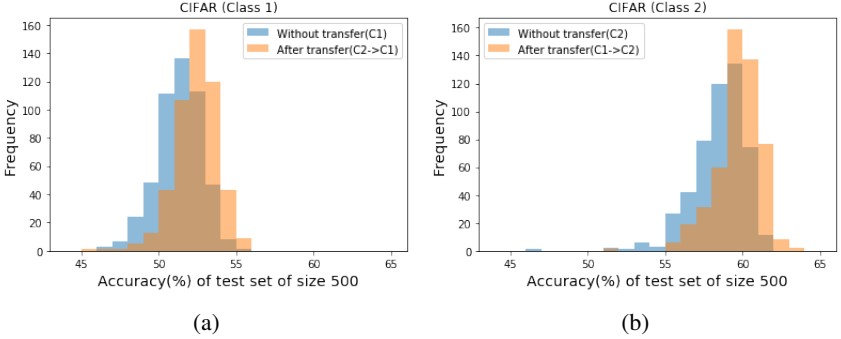

Figure 6: CIFAR classification results. (a) Transfer results for class l. On average, test accuracy increased from 51.28% to 52.42% after transfer. (b) Transfer results for class 2. On average, test accuracy increased from 58.50% to 59.73% after transfer.

Figure 6 shows the results of a positive transfer in the experiment. We conducted the experiment 500 times for each task and obtained the test accuracy. The test accuracy enhanced after a transfer was performed for both classes. For the first class, the total accuracy increased from 51.28% to 52.42% on average; for the second class, the total accuracy increased from 58.50% to 59.73% on average. Furthermore, we verified that the task was learned quickly at the early stage of the training for the post-transferred task compared with the nontransferred task. See Figure 10-(a), (b) in appendix.

## 5 CONCLUSION

We presented a new model called the WBN, which obtains the exact order and correct inputs of function blocks to compose them for constructing target functions. It is especially suitable for reverse engineering, e.g., for revealing PLC devices. It is different from other neural networks as it not only approximates a target function, but also constructs and reveals its structure. The WBN is differentiable with a discretized variable-named selection matrix; therefore, gradient descent can be used for its training. We used the $L_1$ regularization with a variable parameter in temperature learning during training to optimize our network.

Using this network, we discovered that it could reveal LLDs in devices, e.g., logic devices such as PLCs. This network provides the target logic, which may be helpful in auto-reverse engineering for reconstructing PLC-like devices. A WBN provides an equivalent logic rather than an exact one. Furthermore, we discovered that if the layer was set too large, then a WBN may yield an inefficient logic. This will be investigated in future studies.

Additionally, we created another structure to verify the efficacy of the selection layer. Assuming that the function blocks were not known, we created an end-to-end PathNet structure using the selection layers. We performed a few experiments using this network and the transfer learning paradigm; consequently, positive transfer results were shown. This demonstrated that the selection layer could successfully select networks or functions during training.

However, the hyperparameters and their scales must be investigated. The efficient initial regularization parameter and temperature based on the data scale remain unclear. This will be investigated in future studies as it is an important topic.

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

## A APPENDIX

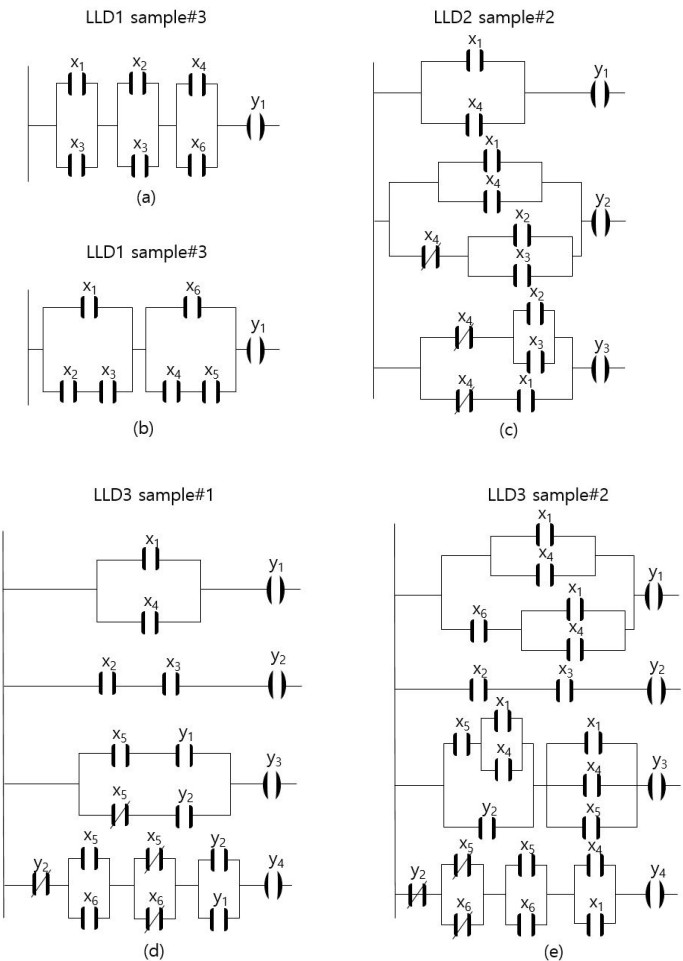

Figure 7: Additional sample structures for three LLDs suggested by WBN. (a), (b) are from LLD1, (c) is from LLD2 and (d), (e) are from LLD3.

```
Expected Equation by WBN is
y1 = (((X4ᵥX6)ᵥ(X6∧X1))∧(X3ᵥ(X1∧X2)))
```
(a)

```
Expected Equation by WBN is
y1 = ((X4∧X4)ᵥ(X1ᵥX4))
y2 = ((X3ᵥX2)ᵥ(X1ᵥX4))
y3 = ((~(X4∧X4))∧((X3ᵥX2)ᵥ(X1ᵥX4)))
```
(b)

```
Expected Equation by WBN is
y1 = ((X4ᵥX1)ᵥ(X4ᵥX1))
y2 = (X2∧(X3∧X2))
y3 = (((X4ᵥX1)∧(X5∧X5))ᵥ((X3∧X2)∧(~(X5∧X5))))
y4 = (((X5ᵥX6)∧(~(X3∧X2)))∧(((~X6)ᵥ(~X5))∧((X3∧X2)ᵥ(X4ᵥX1))))
```
(c)

Figure 8: Sample target functions for three LLDs suggested by WBN. (a) sample equation for LLD1 target function. (b) sample equation for LLD2 target function. (c) sample equation for LLD3 target function

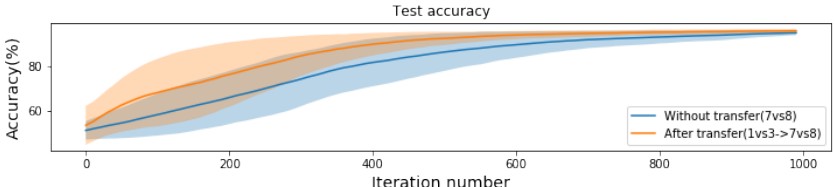

Figure 9: Learning curve for binary MNSIT classification (1vs3 → 7vs8). We conduct experiments 10 times to get the learning curves.

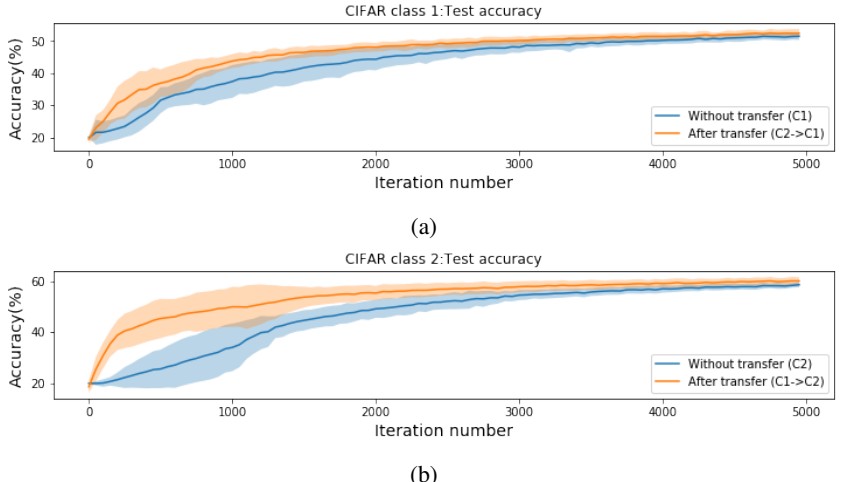

(a)

(b)

Figure 10: Learning curve for CIFAR classification. (a) learning curve for class 2. (b) learning curve for class 1. We conduct experiments 10 times to get the learning curves.

