# OpenReview forum: "White Box Network: Obtaining a right composition ordering of functions"
_ICLR.cc/2020/Conference — Reject_

### Official Review · AnonReviewer3 · 2019-10-16
**Official Blind Review #3**

**Rating:** 1

**Review:**

This paper investigates the question of identifying concise equations from data to understand the functional relations. In particular, a set of base functions are given in hand and the goal is to obtain the right composition of these functions which fits the target function. The main contribution of the paper is to introduce a selection layer, which enhances sparse connections in the network. Several experiments are conducted to show the effectiveness of the method.

My main concern of the paper is about the novelty and the lack of comparison of existing methods. The framework of finding functional relations is set up in [1,2], the main contribution of the paper is a refine architecture with the introduction of the selection layer. However, this selection layer is nothing but incorporating a softmax function. The idea of combining softmax functions in the hidden layers is not novel neither, which could be found in [3,4]. As a result, I find the contribution of the paper very limited, which could be summarized as applying an existing technique on a specific problem. Moreover, in the experimental section, there is a lack of comparison with existing methods such as EQL[1,2] and I consider it a major omission.

Overall, due to the novelty concern and the lack of comparison, I do not support publication of the paper.

[1] Sahoo et al.  Learning Equations for Extrapolation and Control
[2] Martius et al. Extrapolation and learning equations
[3] Graves et al.  Neural turing machines
[4] Graves et al.  Hybrid computing using a neural network with dynamic external memory

**Experience Assessment:**

I have read many papers in this area.

**Review Assessment: Checking Correctness Of Derivations And Theory:**

N/A

**Review Assessment: Checking Correctness Of Experiments:**

I assessed the sensibility of the experiments.

**Review Assessment: Thoroughness In Paper Reading:**

I read the paper thoroughly.

---

> ### Author Response · Authors · 2019-11-10
> **Response to Reviewer 3**
>
> We thank you for your valuable comments.
>
> As you said we combine function block activation with selection layer drawn from softmax layer with temperature. Actually, we tried many different ways including the softmax layer, for example, we tried a power method as NTM [3] does, we also used a probabilistic method like gumbel softmax, and tried RL to learn the path. Among them, softmax with low temperture and softmax with learning temperature give the best performances in our experiments.
> Furthermore, we find out that we have to restrict l1 norm of W (the matrix before going through softmax), otherwise the network often falls deep into a local minimum. Also we use curriculum learning since above restriction can be the obstacle after the network finds a right order. So we reduces the regression constant during the training task and this helps network to find out the right ordering to obtain the function composition.
>
> As for your comments on the experiments, we will try to compare our methods with existing ones such as EQL since those networks also target finding a right equation including extrapolation. We will try to make an experiment comparing with EQLs not only focusing on the error but also on the resulting objective function form. Thank you again for the comments.
>
> [1] Sahoo et al.  Learning Equations for Extrapolation and Control
> [2] Martius et al. Extrapolation and learning equations
> [3] Graves et al.  Neural turing machines
> [4] Graves et al.  Hybrid computing using a neural network with dynamic external memory

---

### Official Review · AnonReviewer2 · 2019-10-22
**Official Blind Review #2**

**Rating:** 1

**Review:**

This paper presents White Box Network (WBN), which allows for composing function blocks from a given set of functions to construct a target function. The main idea is to introduce a selection layer that only selects one element of the previous layer as an input to a function block. This allows for both introducing function priors as well as interpreting the learned function. The paper also presents a setting where each function block is a neural network that can be learned end-to-end using a PathNet style setting and shows positive transfer across MNIST and CIFAR classification tasks.

This presents an interesting technique to enforce learning composition of function blocks while learning the target function. This is important for both interpretability of the learned function as well as for introducing prior information about useful functions in a given domain. The extension to learnable functions (in the form of neural networks) for learning pathways for different tasks is also promising.

The biggest weakness of the paper is that it does not compare both theoretically as well as empirically with several closely related techniques such as RoutingNetworks[1], Modular Networks[2], Neural RAM[3], Compositional Recursive Learner[4], etc. (please find the references below). Routing Networks allow for selecting among a set of function blocks given some inputs. Module Networks similarly introduce modular layer that determines the appropriate modules given the inputs from the previous layers. Neural RAM learns to compose differentiable functions to learn a target function (similar to learning the LLD programs).

It would be good to describe the differences between the proposed approach in WBN and these approaches, as they all seem to propose a similar solution of learning target functions by learning to compose function blocks and reusing the learnt computations for transfer learning. It would also be important to empirically evaluate the related approaches to better understand the pros/cons of WBN compared to these approaches.

What is the biggest size LLD programs that can be learned by WBNs? It would be interesting to evaluate the scalability of the approach to understand the limits of learning complex target functions.

I was also curious if instead of providing the four pre-defined function block (Identity, not, and, or), what would the behavior be if they were all neural networks and also learnt in an end-to-end fashion somewhat similar to [4].

For the MNIST and CIFAR classification tasks, the function blocks are neural networks themselves. After training them using the PathNet like training, is the learned network more interpretable? It might be interesting to see if the selection layer and the function blocks learned some semantic information that might be easier to distill.

For a better comparison with Neural RAM, it might also be interesting to empirically evaluate the performance of WBNs on algorithm induction tasks such as the one used in [3].

[1] Clemens Rosenbaum, Tim Klinger, Matthew Riemer. Routing Networks: Adaptive Selection of Non-Linear Functions for Multi-Task Learning. ICLR 2018

[2] Louis Kirsch, Julius Kunze, David Barber. Modular Networks: Learning to Decompose Neural Computation. NeurIPS 2018

[3] Karol Kurach, Marcin Andrychowicz, Ilya Sutskever. Neural Random-Access Machines. ICLR 2016

[4] Michael Chang, Abhishek Gupta, Sergey Levine, Thomas Griffiths. Automatically Composing Representation Transformations as a Means for Generalization. ICLR 2019

Minor:

page1: ?. -> ?
page 2: questions and answering --> question answering
page 3: y^(l) represents (m(l)+n(l))-dimensional vectors --> should this be m(l) + 2*n(l)?

**Experience Assessment:**

I have published one or two papers in this area.

**Review Assessment: Checking Correctness Of Derivations And Theory:**

I carefully checked the derivations and theory.

**Review Assessment: Checking Correctness Of Experiments:**

I assessed the sensibility of the experiments.

**Review Assessment: Thoroughness In Paper Reading:**

I read the paper thoroughly.

---

> ### Author Response · Authors · 2019-11-10
> **Response to Reviewer 2**
>
> We thank you for the detailed review. We carefully read those four papers you mentioned. As you said, those ideas and networks are related to WBN and we will cite them in the future revision.
>
> We firstly made this network because we aim to solve reverse engineering related works which are usually regression problems. We thought that finding an exact equation for regression task is much more difficult than finding out the order of the task for usual programing for example copying, sorting since for regression, we cannot put an intermediate value during a training session. For example, for image transformation in Compositional Recursive Learner (CRL) paper, we can put intermediate tasks and images to make the network learn how to transform images. The network learns which kind of images have to rotate, resize, and translate since we teach those things during the training. But in regression task (like reverse engineering), we only know inputs and outputs, not intermediate values. So we just tried to find an equation learning network.
>
> The main difference of WBN from others is that this targets for regression task and this network is not doing a multi task learning. For [1], [2], [3] and [4], the order of task differs by inputs. In image transformation in [4], some images have to rotate and some has to resize. This differs image by image so this CRL network is learning how to learn transforming images. But for WBN, this network is just learning the order of the equation not learning how to order equations. So we used much simpler form of network comparing to CRL (CRL has controller outside of model but WBN doesn’t). We want to show that this simple network can find the equation without the intermediate values.
> For LLD programs, we designed the problem for at most 6 composition of basic logic functions. As you suggested, we will check and understand the limits of learning complex target functions.
> We will also try to do an experiment to see if this WBN can learn induction similar to other networks you mentioned if we change WBN into the network with controller. Thank you for the suggestion.
> For MNIST and CIFAR classification tasks, we will try to do the experiment with convolution network and check the whether it learns semantic information.
> Thank you again for pointing out all the details.
>
> [1] Clemens Rosenbaum, Tim Klinger, Matthew Riemer. Routing Networks: Adaptive Selection of Non-Linear Functions for Multi-Task Learning. ICLR 2018
>
> [2] Louis Kirsch, Julius Kunze, David Barber. Modular Networks: Learning to Decompose Neural Computation. NeurIPS 2018
>
> [3] Karol Kurach, Marcin Andrychowicz, Ilya Sutskever. Neural Random-Access Machines. ICLR 2016
>
> [4] Michael Chang, Abhishek Gupta, Sergey Levine, Thomas Griffiths. Automatically Composing Representation Transformations as a Means for Generalization. ICLR 2019

---

### Official Review · AnonReviewer1 · 2019-10-23
**Official Blind Review #1**

**Rating:** 1

**Review:**

My understanding of this paper is that it proposes a combination of simple logical blocks that can efficiently learn logic rules implemented by a logic function. I think the authors define interpretability as the possibility to exactly express an unknown function in a composition of blocks, but despite several reading of this paper, I am not sure. A strong assumption in this paper seems to be that the target function of a supervised task can be exactly expressed via some compositional blocks. I would suggest a significant revision to clearly explain why WBNs are more interpretable and avoid any vague terminology.

I will list below some of my concerns:

- After reading several times this paper, I do not understand why this method is "more interpretable". The explanation of the papers are quite verbose. I tried to phrase my concern as this: could a standard CNN be more interpretable than this WBN because it simply uses linear operation? I'd like to see the reaction of the authors to this questions

For instance:
"This differs from the normal neural network because the WBN reveals the exact functions with the correct inputs and their ordering to construct the target function, instead of merely approximating them."
If the target function is precisely a cascade of linear operators and ReLU, then the objective of learning would be to recover exactly the linear operators and wouldn't consist in an approximation. Are the authors trying to tackle the nature of the objective functions? It is very unclear to me.

"It is different from other neural networks as it not only approximates a target function, but also constructs and reveals its structure."
I do not understand why the structure is less opaque than in standard CNNs or how it is revealed.

The authors must clearly define the notion of interpretability in the text, in an explicit and simple manner. As an active researcher in this field, I believe that this is quite difficult because everybody has its own interpretation of interpretability. Here, I would suggest to significantly rephrase this.

- I am quite confused in the notation.. For instance, sometimes \hat W is used, sometimes W...

- It is claimed that $\ell^1$ minimisation will allow to avoid... sparse and sharp operators: "This causes Wˆ (l) to be extremely sparse and sharp, and it can be an obstacle for shifting the function blocks from one ordering to another." This goes against my intuitions/knowledge, could the authors point me to a reference?

- Table 1: Do the mean square errors indicate an exact learning? If yes, this should be commented. Also, the Table is not discussed in the text...

- I do not understand why the CIFAR and MNIST experiments are relevant to this paper. Furthermore, the accuracy are very low.

**Experience Assessment:**

I have published in this field for several years.

**Review Assessment: Checking Correctness Of Derivations And Theory:**

I carefully checked the derivations and theory.

**Review Assessment: Checking Correctness Of Experiments:**

I carefully checked the experiments.

**Review Assessment: Thoroughness In Paper Reading:**

I read the paper thoroughly.

---

> ### Author Response · Authors · 2019-11-10
> **Response to Reviewer 1**
>
> We thank you for your valuable comments.
>
> - As you said, WBN is not necessarily the best way to find out an objective function. If we don’t know any information about the objective function, it is natural to use a linear operator and an activation as deep learning shows a great success in many fields.
> However, in our paper, we have a given assumption that the objective functions which consist of a composition of several known basic functions. So we consider prior information about useful functions in a given domain. We mean by interpretability is that WBN can reveal the exact order of a composition of such basic functions.
> This situation arises in some applications, for example, in a ladder logic diagram field, we know there are some basic logic functions that forms a logic diagram. There are many networks that give a right output with inputs, yet we not only want to find correct outputs but also to construct an objective function with basic logic functions among a given set of such basic functions so we can draw a logic diagram of the objective function.
> According to your comment, we will clearly define the notion of interpretability in the text. We meant in the paper that WBN is more interpretable because it uses prior information about basic functions in a given domain and finds out the right ordering of basic functions so we can reconstruct the objective function with reasonable basic functions in that field.
>
> - We use both (\hat W) and W because we have to distinguish them. W is just a normal weight matrix and (\hat W) is a sharpened one. We make selection layer (\hat W) by sharpening W with softmax.
>
> - Since (\hat W) already went through a softmax function (with a low temperature) so it is already sparse and l1 norm is fixed. So we use regularization with l1 norm of W. What I mean sparse is sparsity of (\hat W); if W_1= [0, 10] and W_2=[-10, 10], W_2 is likely to be more sparse after it goes through the softmax layer. If there are no regularizing terms, the element of W became extremely large. Suppose W1=[1, 10] and W_2= [1, 10000], after softmax layer, both (\hat W1), (\hat W2) approximate [0, 1]. However, the network cannot change (\hat W2) from [0, 1] to [1, 0] after learning even if it has to be with gradient descent. We found that the element of W2=[1, 10000] is too large and this means it often fell deep into a local minimum. To prevent this problem, we give l1 restriction with W matrix. (Actually it doesn’t have to be l1 but we chose it with experimental results.)
>
> - Actually mean square error doesn’t indicate an exact learning but it shows the error can goes smaller with the prior information of given functions. We will put the explanation in the text.
>
> - We consider the case ‘What if we don’t have any information about basic functions in a given field’, and we also thought that a linear function with activation is most natural as basic functions. And after that we have to figure out that WBN choose a right order of linear activation layers. To evaluate the result, we did the same experiment with Pathnet[1]. In that paper, they conducted an experiment to figure out how the network finds a right path for transfer learning. We also do the same experiment to find out whether WBN finds a right order of the layers using a transfer learning method. For the performance, this network is not designed for giving good results for CIFAR and MNIST classification tasks. As a network, it is just a fully connected network with 3 layers. We only aims to know if WBN transfers information well or not. (Actually [1] gives even worse results in terms of the performance). Thank you again for the comments.
>
> [1] Fernando et al, PathNet: Evolution Channels Gradient Descent in Super Neural Network

---

> > ### Comment · AnonReviewer1 · 2019-11-11
> > **Thanks for the feedback**
> >
> > Dear authors,
> >
> > Thanks four feedback. I answer to each of your point below.
> >
> > - Thanks for the clarification, I understand better the purpose of this paper, but then, it means that the 2 first pages must be significant rewritten. I think the concept that you aim to solve is "tractability" rather than "interpretability". In other words, you found a method to recover exactly the parameters of a specific and constraint class of NN. (feel free to correct me if this is not exactly your purpose explained in few words) Again, there is a significant rewriting required.
> >
> > - I think I guessed this, but please have a look to the 3 first paragraphs of Section 3.1 . For instance, "To make $\hat Wˆ (l)$ differentiable, we approximate $\hat Wˆ (l)$ by"... One of those matrix should be $W_l$ and not $\hat W_l$. $W_l$ is neither discussed at all in the text (I did a ctrl+F to make sure I am not mistaking), nor introduced. You never write $W(l)$ corresponds to the weights of the $l$-th layer. It could help a reader to understand the issueslinked to discretization.
> >
> > - OK - I think I understand this is a way to tackle discretization issues. Yet in the text is written that an $l^1$ penalty will help the coefficients to be less sparse, whereas it should be, to my understanding, the opposite.
> >
> > - Is it discussed in the text?(ie, can you give me a line number) I thought Table 1 was computed on each element of the dataset(which would have been huge, I agree) I also thought that if the numerical error is under the numerical precision, one might have some specific comments?
> >
> > - Given the claim of the paper, does it mean you expect the classification function to be tractable? I agree [1] gives different results.

---

> > > ### Author Response · Authors · 2019-11-11
> > > **Thank you for your helpful comments!**
> > >
> > > Thank you for your helpful comments! I answer to each of your point just as before.
> > >
> > > -	We think that it would be good to write more clearly the purpose of this paper.
> > >
> > > -	We only write the expression W in an equation on page 3. As you said we have to introduce earlier W which turns to $\hat{W}$ after discretization.
> > > -	As you mentioned l1 penalty is used to tackle discretization in the early stage of the training. If we do not use l1panalty, the network can fall into a local minimum in the early stage. However, this penalty can be an obstacle since WBN has to be discretized during the training. So we reduces the regression constant during the training task.
> > >
> > > -	We only mentioned about the comparing loss on page 6. That is, “Although comparing losses is not meaningful as our model not only reduces the error, but also obtains the actual structure of the target function, it clearly shows how our model effectively reduces the loss after obtaining the structure of the target function”. We did not discuss about the results on Table 1 because we thought it was not meaningful.
> > >
> > > -	We did this experiment to find out whether the selection layer ($\hat{W}$) can catch a useful function or not. In [1], they reuse the layer selectively which are useful for transfer learning. We were not sure that the classification function is tractable since we don’t know the basic functions. We just use linear layers including pre-trained layers as basic functions. We will check whether those layers can learn semantic information or not later on.

---

> > > > ### Comment · AnonReviewer1 · 2019-11-15
> > > > **Thanks for your reply**
> > > >
> > > > Dear authors,
> > > >
> > > > - Yes.
> > > > - Yes.
> > > > - Alright. This is pretty standard(I think a citation to explain this would be welcome)
> > > > - This was a bit vague, each tables/figure should be discussed in the main text.
> > > > - I think it would be good to include those results in the future paper.

---

### Decision · Program_Chairs · 2019-12-19

**Decision:**

Reject

**Comment:**

This paper presents White Box Network (WBN), which allows for composing function blocks from a given set of functions to construct a target function. The main idea is to introduce a selection layer that only selects one element of the previous layer as an input to a function block.  The reviewers were unanimous in their opinion that the paper is not suitable for publication at ICLR in its current form.  There were significant concerns about the clarity in writing, and reviewers have provided detailed discussion should the authors wish to improve the paper.